# Characterisation of Bioactive Peptides from Red Alga *Gracilariopsis chorda*

**DOI:** 10.3390/md21010049

**Published:** 2023-01-11

**Authors:** Martin Alain Mune Mune, Yoshikatsu Miyabe, Takeshi Shimizu, Wataru Matsui, Yuya Kumagai, Hideki Kishimura

**Affiliations:** 1Faculty of Science, University of Maroua, Maroua P.O. Box 814, Cameroon; 2Chair of Marine Chemical Resource Development, Graduate School of Fisheries Sciences, Hokkaido University, Hakodate 041-8611, Hokkaido, Japan; 3Aomori Prefectural Industrial Technology Research Center, Hachinohe 031-0831, Aomori, Japan; 4Hokkaido Industrial Technology Center, Department of Research and Development, Hakodate 041-0801, Hokkaido, Japan; 5Laboratory of Marine Chemical Resource Development, Faculty of Fisheries Sciences, Hokkaido University, Hakodate 041-8611, Hokkaido, Japan

**Keywords:** red alga, *Gracilariopsis chorda*, bioactive peptides, ACE inhibitory activity, DPP-IV inhibitory activity, DPPH scavenging activity

## Abstract

In this study, we studied the bioactive peptides produced by thermolysin hydrolysis of a water-soluble protein (WSP) from the red alga *Gracilariopsis chorda*, whose major components are phycobiliproteins and Ribulose-1,5-bisphosphate carboxylase-oxygenase (RuBisCo). The results showed that WSP hydrolysate exhibited significantly higher ACE inhibitory activity (92% inhibition) compared to DPP-IV inhibitory activity and DPPH scavenging activity. The phycobiliproteins and RuBisCo of *G. chorda* contain a high proportion of hydrophobic (31.0–46.5%) and aromatic (5.1–46.5%) amino acid residues, which was considered suitable for the formation of peptides with strong ACE inhibitory activity. Therefore, we searched for peptides with strong ACE inhibitory activity and identified two novel peptides (IDHY and LVVER). Then, their interaction with human ACE was evaluated by molecular docking, and IDHY was found to be a promising inhibitor. In silico analysis was then performed on the structural factors affecting ACE inhibitory peptide release, using the predicted 3D structures of phycobiliproteins and RuBisCo. The results showed that most of the ACE inhibitory peptides are located in the highly solvent accessible α-helix. Therefore, it was suggested that *G. chorda* is a good source of bioactive peptides, especially ACE-inhibitory peptides.

## 1. Introduction

Red algae are commonly used by the people near the coastal areas in Asia and Europe as a food source, to prepare many kind of dishes and soup. The consumption of red algae is actually expanding because they are a good source of essential nutrients, minerals and vitamins, and also contain bioactive compounds with attractive biological activities. The compounds found in red algae such as amino acids, proteins, polysaccharides, polyunsaturated fatty acids, sterols and polyphenols exhibited antioxidant, antimicrobial, antihelmintic, antidiabetic, antihypertensive, antiinflammatory and anticoagulant properties [1]. Applications of these bioactive compounds in the pharmaceutical industry are now very common. In this regard, bioactive metabolites found in algae are used to treat AIDS (acquired immune-deficiency syndrome), inflammation, arthritis, and microbial infections [2]. The red algae of Gracilariaceae were found particularly attractive because they presented the ability to achieve high yields and produced extracts with important industrial and biotechnological applications. In this regard, *Gracilariopsis chorda*, one of the most popular red algae of Gracilariaceae, is utilized as a functional food in Korea, and as a raw material in Europe and Asia for the production of agar [1,2]. Red alga generally contains a high quantity of proteins, mainly located on the stromal side of thylakoid membranes in the chloroplast, arranged as phycobilisomes and called phycobiliproteins, as well as ribulose-1,5-bisphosphate carboxylase-oxygenase (RuBisCo). These water-soluble proteins (WSP) particularly contain sequences of many potent bioactive peptides encrypted in their polypeptide sequence. Studies with red algae revealed the presence of peptides with Angiotensin-converting enzyme (ACE), dipeptidyl peptidase-IV (DPP-IV) inhibitory activity, as well as antioxidant properties in the sequence of phycobiliproteins, which could be released following enzymatic hydrolysis [3,4,5,6,7,8]. Therefore, it could be important to enhance utilization of *G. chorda* as a functional food by the release of potential bioactive peptides found in the sequence of WSP. Such an ingredient could find application in the prevention or treatment of diseases such as diabetes and hypertension, since synthetics drugs produce many undesirable side effects. However, to be able to optimize and predict the release of bioactive peptides from chloroplast proteins by enzymatic hydrolysis, their sequences are usually required, as well as 3D conformation.

The main WSP found in red algae are phycobiliproteins and RuBisCo. These proteins play an important role during the photosynthesis. Basically, phycobiliproteins contain phycoerythrin (PE), phycocyanin (PC) and allophycocyanin (APC), which are covalently linked to one or several chromophores at specific Cys residues. In a previous study, we analyzed the steric structure of PE from dulse (*Devaleraea inkyureii*), which is a type of red algae harvested in the cold waters of Japan. We found that its structure is an α-helix-rich (αβ)_6_ hexamer complex with a toroidal shape [7]. It is well known that enzymatic hydrolysis of chloroplast proteins from red algae produces hydrolysates with important ACE inhibitory, DPP-IV inhibitory, and antioxidant activities. Many potent bioactive peptides have been purified from these hydrolysates. Particularly, new ACE inhibitory peptides were purified from algae WSP hydrolyzed by thermolysin [9], and inhibition mechanism was found competitive and non-competitive [9,10]. Recently, the analysis of the whole chloroplast genome from dulse was used as a tool for a wide characterization of bioactive peptides in the sequence of chloroplast proteins [3]. Presence of bioactive peptides with a wide range of bioactivities has been revealed. However, interaction between protease with chloroplast proteins also depends on the conformation of the protein in the mixture. It is well known that hydrolysis of protein secondary structures such as sheets and hairpins is more difficult than alpha helix conformation or turns [11]. Therefore, to be able to design efficient processes for the hydrolysis of chloroplast proteins to produce bioactive peptides, it is important to analyze the release of bioactive peptides from the protein matrices. The complex rod-shape structure of phycobiliproteins rich in alpha-helix secondary conformation, as well as the 3D structure of RuBisCo [12], probably affects the release of bioactive peptides encrypted in their primary sequence. Actually, bioinformatic tools are increasingly used to predict the structure of proteins as well as the release of bioactive peptides, with great accuracy. In this regard, combining information from the chloroplast protein’s structure and bioactive properties of protein hydrolysates could be important to understand structural factors that influence the release of bioactive peptides from red algae chloroplast proteins. Therefore, in this study *G. chorda* soluble chloroplast proteins were hydrolyzed by thermolysin, then ACE inhibitory, DPP-IV inhibitory, and antioxidant activities of the hydrolysate were determined. Moreover, protein hydrolysate was fractionated, and the release of bioactive peptides related to ACE inhibitory effect was analysed by several in silico techniques to understand the structure–function relationship of *G. chorda* WSP.

## 2. Results and Discussion

### 2.1. Characteristics of G. cholda WSP

WSP from red alga usually contain a high amount of phycobiliproteins, which are composed primary of α and β subunits, each linked covalently to one or several phycobilin chromophores at specific Cys residues [13]. The linked Cys residues are usually well conserved in the primary structure of phycobiliproteins, and in *G. chorda* (Figure 1a), their linkage with phycobilin chromophore provided a characteristic red color to WSP (Figure 1b). The absorption spectrum of *G. chorda* WSP is characteristic to the chromophore composition in the mixture (Figure 1c). WSP presented high absorbance at 543, 565 and 500 nm, and low absorbance at 620 and 655 nm. Generally, PE presents maximum absorbance at λ = 490–570 nm, PC at λ = 610–625 nm and APC at λ = 650–660 nm. PE binds phycoerythrobilin and phycourobilin chromophores, PC binds phycoerythrobilin and phycocyanobilin, and APC binds phycocyanobilin. Therefore, the high absorbance at 500–565 nm of *G. chorda* soluble protein is characteristic to high content of PE in the mixture. Similar results were obtained for other red algae such as dulse [3]: i.e., we calculated the ratio of PE, PC, and APC present based on the absorbance of the visible light absorption spectrum, and the main component of the phycobiliprotein was PE (PE = 5.0 mg/mL, PC = 1.8 mg/mL, APC = 0.01 mg/mL) [14]. The electrophoresis analysis of *G. chorda* WSP provided additional information on the protein composition (Figure 1d). It was observed by SDS-PAGE analysis two major bands at approximately 20 and 55 kDa, and two minor bands at 16 and 32 kDa. The band at 55 kDa was also observed in WSP from Japanese dulse and *P. pseudolinearis* and was related to the large subunit of RuBisCo (RuBisCo-L) [3,13]. RuBisCo is a multimeric protein which plays an important role in the assimilation of CO_2_ to form sugars during photosynthesis, and then one of the most abundant WSP in plants. The band at 16 kDa was probably related to the small subunit of RuBisCo (RuBisCo-S). The band at 20 kDa produced fluorescence following excitation at 490–560 nm, indicating it was phycobiliproteins linked to chromophores.

### 2.2. Bioactive Properties of G. chorda WSP Hydrolysate and Isolation of ACE Inhibitory Peptides

Previously, hydrolysis of red algal proteins produced peptides with several types of bioactivities [3,5,6,7]. In this study, *G. chorda* WSP were hydrolyzed by thermolysin, and bioactive properties of the hydrolysate were evaluated, especially ACE inhibitory activity, DPP-IV inhibitory activity, and DPPH scavenging activity (Figure 2a–c). It was observed as a significant increase in these activities in the hydrolysate compared to WSP. The particularly high ACE inhibitory activity of *G. chorda* hydrolysate (92%) compared to WSP (35%) was noteworthy. Conversely, *G. chorda* WSP thermolysin hydrolysate was a poor source of DPP-IV inhibitory (28% inhibitory activity at 5.0 mg/mL) and DPPH radical scavenging peptides (63% DPPH scavenging activity at 3.2 mg). Proteins from the red alga dulse previously showed low radical scavenging activity, probably due to low content of amino acids capable of scavenging free radicals [6]. In addition, high ACE inhibitory activity was also observed in red alga *Porphyra dioica* proteins hydrolyzed by Alcalase and Flavourzyme [15], and DPP-IV inhibitory effect was noticed in Irish dulse proteins digested by Corolase PP [16]. In a study of Japanese dulse, it was reported that the antioxidant activity of WSP (DPPH and ABTS radical scavenging activity) is relatively low and that the main antioxidant activity of WSP is derived from the chromophore of phycobiliproteins [6]. Therefore, we believe that the antioxidant activity of WSP and its hydrolysates is low, even when antioxidant activity is measured by other methods.

Since *G. chorda* WSP hydrolyzed by thermolysin presented interesting potential for utilization as functional food in the prevention or treatment of hypertension, and then subjected to fractionation by RP-HPLC and mass spectrometry to identify bioactive peptides (Figure 3a,b). For this purpose, four fractions (Fractions 1, 2, 7, and 8 in Figure 3) with ACE inhibitory activity > 40% were selected for peptide identification. The mass spectrometry analysis of fractions 2, 7, and 8 provided 3 peptide sequences, respectively, IDHY, LVVER, and LRY. No peak with m/z ratio > 400 was identified in fraction 1. This fraction probably contained a dipeptide with the m/z peak mixed with those representing the matrix used for MALDI during mass spectrometry. Thermolysin is an endopeptidase which recognises amino groups of a wide range of hydrophobic amino acid residues, including Ile, Leu, Val and Phe. Several potent dipeptides with hydrophobic amino acid residues at the amino terminal, such as VY, IY and IW, have been identified in plant protein hydrolysates prepared by thermolysin [17,18]. The three peptides LRY, IDHY, and LVVER were checked in the BIOPEP-UWM database (https://biochemia.uwm.edu.pl/biopep/start_biopep.php), and the result showed that IDHY and LVVER were new peptides not yet registered. The two new peptides exhibited some important characteristics for ACE inhibition such as presence of Arg and Tyr at the C-terminal residue [19]. However, IDHY was probably more potent than LVVER for the inhibition of ACE, since it showed higher score in PeptideRanker (Table 1). The peptide LRY was already found in thermolysin hydrolysates of other red algal WSP [3,20]. In the previous study we identified a novel ACE inhibitory peptide LRY from dulse [2]. The IC_50_ value of synthesized LRY against the ACE from rabbit lung was 0.044 μmol, and its effect was the same level as that of the sesame peptide LVY (IC_50_ value: 0.045 μmol) used as a Food for Specified Health Uses (FOSHU) in Japan. Then, we performed a docking simulation of LRY with human ACE and found that −9.50 kcal/mol. Since the affinity of IDHY for human ACE in this study was −9.50 kcal/mol, it was presumed that IDHY has a hypotensive effect on humans as well as LVY. We are currently planning a clinical trial on the blood pressure-lowering effects of LRY and IDHY from red algae in collaboration with a company.

### 2.3. Structural Characterization of G. chorda WSP and Structure-Function Relationship

To optimize the application of *G. chorda* WSP in the preparation of hydrolysate with high ACE inhibitory activity by in silico analysis, it is important to elucidate the structure of WSP. However, *G. chorda* contained relatively high amounts of polysaccharides. In addition, the phycobiliproteins are very hydrophobic and adsorb to the column resin, resulting in very low yields. For these reasons, it was very difficult to isolate and purify the major proteins of WSP. Therefore, we needed to approach the elucidation of their primary structure by genetic analysis. Since the genes of phycobiliproteins and RuBisCo, major proteins of WSP, are encoded in chloroplast DNA, as in previous studies [4,20], we determined the complete chloroplast genome and manually annotated it. The genes of the main WSP were found, and their amino acid sequences were deduced (Appendix A, Appendix A, AP017366 in NCBI). It was noteworthy that PE possessed a gamma subunit (PE-γ) in addition to α and β subunits (PE-α, PE-β), i.e., we obtained 28,573 contigs by next generation sequencing of *G. chorda* DNA and we found the contig (LC713007 in NCBI) containing PE-γ gene encoded in nuclear DNA. The α and β subunits of PE construct (αβ)_6_ as described above. On the other hand, PE-γ is placed in the central space of the (αβ)_6_ hexamer complex as linker protein to form part of the rod of the phycobilisome. Compared to the α and β subunits, there have been fewer structural and functional studies of the γ subunits, because the gene is encoded in the nucleus [21]. APC also presented three subunits, namely α, β and γ (APC-α, APC-β, and APC-γ). PC α and β subunits (PC-α, PC-β) and RuBisCo-L and -S genes were also encoded in *G. chorda* chloroplast genome. From the deduced amino acid sequences, the amino acid compositions of these proteins were analysed. High percentages of total hydrophobic amino acids (31.0–46.5%) and total aromatic amino acids (5.1–10.1%) were observed in *G. chorda* phycobiliproteins and RuBisCo (Table 1). In addition, Pro was found at 1.7–4.1%, and Trp was particularly abundant in RuBisCo-L (1.6%). High hydrophobic amino acid content was also found in Japanese dulse chloroplast proteins [4]. The important role of bulky hydrophobic amino acids (Pro, Phe, Tyr, and Trp) at the C-terminal, and the N-terminal aliphatic amino acid for di- and tripeptides for high ACE inhibitory activity is well known [19]. Moreover, the calculated molecular weight of WSP on the basis of their amino acid composition was found in accordance with SDS-PAGE analysis, and the calculated pI confirmed their water solubility (Appendix A, Appendix A).

The origin of detected ACE inhibitory peptides was determined by mapping their sequences in the primary sequence of WSP (Appendix A). It was found that the sequences of IDHY, LRY and LVVER were, respectively, detected in the primary structure of PEα, PEβ and RuBisCo-L, in a single occurrence. Presence of other potential ACE inhibitory peptides after hydrolysis of *G. chorda* WSP by thermolysin was predicted using the BIOPEP tool. Five peptides with IC_50_ ≤ 100 µM were detected and resulting from the primary structure of PE-α, PE-β, PC-α, PC-β, APC-α, APC-β, APC-γ, and RuBisCo-L (Appendix A).

In addition, it was observed that the peptides IW and IK with lower IC_50_ were found in the primary structure of RuBisCo-L. Afterwards, the 3D structure of *G. chorda* WSP was predicted in silico to understand the structural factors that contributed to the release of ACE inhibitory peptides. The modeled structures are then presented in Figure 4a–d and their secondary structure was analysed by PDBsum (Table 2). It was observed that phycobiliproteins were rich in alpha helix (7 to 14) secondary structure, and also contained beta turns (2 to 13) and gamma turns. PE-**γ** was the more complex phycobiliprotein with 14 helices (62.1%), 13 beta turns and 1 gamma turn (Table 2). This secondary structure is common to phycobiliproteins in many red algae species and the high number of alpha helices is essential for the building of phycobilisomes rod-shape structure [12]. The secondary structure of RuBisCo was more complex. RuBisCo-L contained 25 alpha helices and 27 beta turns, but also 15 strands and 2 beta hairpins. RuBisCo-S also contained 4 strands and 2 beta hairpins, in addition to alpha helix (3) and beta turns (5). Generally, the sheet secondary structure is less susceptible to enzymatic hydrolysis than the helix one, and presence of beta hairpins conferred additional stability of the structure [11].

Peptides with ACE inhibitory activity were further mapped on the secondary structure of *G. chorda* WSP to analyze their structural environment in the protein (Figure 4 and Table 1). It was found that all the detected and potential peptides were located in the helix region of different proteins, except for the peptide VK which is located in sheet. Solvent accessible surface (SAS) is an additional tool to analyze the accessibility of peptides encrypted in the protein structure to proteases. Generally, the region with SAS < 0.3 is buried and those with SAS > 0.3 are exposed [22]. Then, as expected, all the detected peptides were located in exposed regions. Potential ACE inhibitory peptides were also exposed to the solvent, excepted for the peptide AR located in PE-α. Solubility of the peptides was finally predicted to evaluate recovery of the released peptides in water. It was found that all the detected peptides were soluble in water. In contrary, peptides IW and FQ were less soluble in water.

### 2.4. Prediction of the Interaction between New Bioactive Peptides and Human ACE

To confirm the potential of a new peptide to inhibit ACE, generally in vitro or in vivo experiments are required. However, recent simulation of the interaction between ACE and different peptides by using in silico methods gave good correlation with results obtained in vitro. In addition, the *in silico* methods produced an overview of the interaction at the atomic level [23]. This is convenient to make a decision. Therefore, inhibition of human ACE by IDHY and LVVER was evaluated by molecular docking, and results were compared to those of lisinopril and captopril (Table 3 and Figure 5). Docking methodology was valid because RMSD was less than 1 Å. The affinity energy between ACE and the peptides was lower for IDHY (−9.5 kcal/mol) compared to LVVER (−6.9 kcal/mol). As a result, IDHY interacted with ACE through 7 hydrogen bonds and 2 bonds with Zn^2+^ ion, while ACE interacted with LVVER through 6 hydrogen bonds. Zn^2+^ ion plays an important role in the activity of ACE because it facilitates the binding of the substrate to the Zn^2+^-binding motif HEXXH at the active site which includes His383, His387 and Glu411. In addition, Zn^2+^ ion assists the nucleophilic attack from an activated water molecule, with the reaction being initiated by the carboxyl group of Glu384 [23]. Therefore, potent ACE inhibitors should bind to specific amino acid residues at the active site, together with coordinating Zn^2+^, in order to prevent the fixation of the substrate to the active site. In this regard, IDHY coordinated with the Zn^2+^ ion and also interacted with Glu384 through a hydrogen bond of 2.926 Å. Moreover, IDHY was linked to His387 and Glu411 by salt bridges (Figure 5). No interaction was observed between the specific amino acid residues at the active site and LVVER. Furthermore, the number of non-bonded contact was also important, because interaction of inhibitor with a high number of amino acid residues near the active site could alter the conformation structure and then prevent the fixation of the substrate. On the other hand, Lisinopril interacted with ACE through coordination bonds with Zn^2+^ ions, hydrogen bonds and non-bonded interactions with specific residues at the active site, as well as those near the active site (Figure 5c).

## 3. Materials and Methodsw

### 3.1. Materials

Samples (*G. chorda*) were collected on the coast of Hakodate, Hokkaido Prefecture, Japan. Hexadecyltrimethylammonium Bromide (CTAB), tris-[hydroxymethyl]amino-methane (Tris), ethylenediamine-*N*,*N*,*N*′,*N*′-tetraacetic acid, disodium salt, dihydrate (EDTA), Proteinase K (EC 3.4.21.64, from *Tritirachium album*), RNase A (EC 3.1.27.5, from bovine pancreas), porcine stomach pepsin (EC 3.4.23.1) and bovine pancreatic trypsin (EC 3.1.21.4) were purchased from FUJIFILM Wako Pure Chemical (Osaka, Japan). Phenol-chloroform-isoamyl alcohol (25:24:1) was purchased from NACALAI TESQUE, INC (Kyoto, Japan). Ala-Pro-*p*-nitroanilide (Ala-Pro-*p*NA) were also obtained from Bachem AG (Bubendorf, Switzerland). All other regents were purchased from FUJIFILM Wako Pure Chemical (Osaka, Japan).

### 3.2. Preparation of G. chorda WSP Hydrolysate

The WSP from *G. chorda* were prepared according to the same method as in our previous paper [3]: i.e., frozen *G. chorda* was lyophilized and ground into a fine powder. To this powder, 20 v/w distilled water was added and proteins were extracted at 4 °C for 7 h. The extract was centrifuged at 4 °C, 15,000× *g* for 10 min, and then the supernatant was used as *G. chorda* WSP. The visible ray absorption spectrum of *G. chorda* WSP was analyzed by a spectrophotometer (UV-1800, Shimadzu, Kyoto, Japan). *G. chorda* WSP hydrolysate were prepared as previously reported [3,13]. The WSP were hydrolyzed by 1.0 wt% of thermolysin at 70 °C for 3 h, and the reaction was ended by heat treatment at 100 °C for 5 min. Subsequently, the solution was centrifuged at 4 °C, 15,000× *g* for 10 min. The supernatant was dried by lyophilisation into *G. chorda* peptides.

For analysis of protein composition, sodium dodecyl sulfate-polyacrylamide gel electrophoresis (SDS-PAGE) was carried out using a 0.1% SDS-13.75% polyacrylamide slab-gel by the method of Laemmli [24]. The gel was stained with 0.1% Coomassie Brilliant Blue R-250 in 50% methanol-7% acetic acid and the background of the gel was destained with 7% acetic acid. Fluorescence of phycobiliprotein on slab-gel was detected by gel documentation LED illuminator (VISIRAYS AE-6935GN: ATTO, Tokyo, Japan).

### 3.3. ACE Inhibitory Assay

ACE inhibitory assay was carried out according to the method of Cheng and Cushman [25], with some modifications. Fifteen microliters of sample solution (3.2 mg/mL) were added to 30 µL of ACE (0.2 units/mL), and the mixture was pre-incubated at 37 °C for 5 min. Thirty microliters of Hip-His-Leu solution (12.5 mM in 0.1 M sodium borate buffer containing 400 mM NaCl at pH 8.3) were added to the mixture. After incubation at 37 °C for 1 h, the reaction was stopped by adding 75 µL of 1.0 M HCl. The released hippuric acid was extracted with 450 µL of ethyl acetate. Four hundred microliters of the upper layer were evaporated, and then the hippuric acid was dissolved in 1.5 mL of distilled water. The absorbance at 228 nm of the solution was measured by a spectrophotometer. The inhibition was calculated from the following equation: [1 − (As − Asb)/(Ac − Acb)] × 100, where Ac is the absorbance of the buffer, Acb is the absorbance when the stop solution was added to the buffer before the reaction, As is the absorbance of the sample, and Asb is the absorbance when the stop solution was added to the sample before the reaction.

### 3.4. DPP-IV Inhibitory Assay

Preparation of the recombinant dipeptidyl peptidase-IV (DPP-IV, EC 3.4.14.5) from human kidney was carried out according to the method of Hatanaka et al. [26]. DPP-IV inhibitory activities with *G. chorda* protein and peptides were analysed by a slight modification of the method of Hatanaka et al. [26]: dried samples were dissolved with 20 mM Tris-HCl (pH 7.5) (16.6 mg/mL). Thirty micro liters of each sample (0.5 mg, final concentration of 5 mg/mL) was added to 20 μL (0.029 U, final concentration of 0.29 U/mL) of DPP-IV in 20 mM Tris-HCl (pH 7.5), and then the mixture was pre-incubated at 37 °C for 3 min. The enzymatic reaction was initiated by adding 50 μL (final concentration of 0.5 mg/mL) of 1.0 mg/mL Ala-Pro-*p*NA in 20 mM Tris-HCl (pH 7.5). This mixture was incubated at 37 °C for 10 min and measured the increased absorbance at 405 nm using a UV-1800 spectrophotometer (Shimazu, Kyoto, Japan). One unit (U) of the enzyme activity was defined as the amount of enzyme that liberates 1 μmol *p*-nitroaniline per min under the assay conditions. The percentage of inhibition was determined relative to the enzyme activity without samples.

### 3.5. DPPH Radical Scavenging Assay

DPPH radical scavenging assay was carried out according to the method of Sharma and Bhat [27] with some modifications. Sample or distilled water were mixed with 1.0 mL of 500 µM DPPH solution (in 99.5% ethanol) and 800 µL of 100 mM Tris-HCl buffer (pH 8.0). The mixture was incubated at room temperature in dark condition for 20 min. After the incubation, the solution was centrifuged (3500, KUBOTA, Osaka, Japan) at 4 °C, 2000× *g* for 10 min, and the supernatant was measured absorbance at 517 nm. DPPH radical scavenging activity was calculated from the following equation: [1 − (As − Asb)/Aw] × 100, where As is the absorbance of sample mixed with DPPH solution and Asb is the absorbance of sample mixed with ethanol, and Aw is the absorbance of distilled water mixed with DPPH solution. All the assays were performed in triplicate.

### 3.6. Separation of G. chorda WSP Hydrolysate

The *G. chorda* hydrolysate was dissolved in ultrapure water containing 0.1% trifluoroacetic acid (TFA) and applied to sequential filtration by Millex-GV (pore size: 0.22 µm) and Millex-LG (pore size: 0.20 µm). Peptides in the filtrate were isolated by reversed phase-HPLC (RP-HPLC) with a Mightysil RP-18GP column (4.6 × 150 mm) (Kanto Kagaku, Tokyo, Japan) using a linear gradient of acetonitrile (1–20%) containing 0.1% TFA at a flow rate of 1.0 mL/min.

The amino acid sequences of the ACE inhibitory peptides were analysed by MALDI-TOF/MS/MS using a 4700 Proteomics Analyser mass spectrometer with *DeNovo* Explorer ver. 3.6 (Applied Biosystems, CA, USA).

### 3.7. Statistical Analysis

All experiments were replicated at least three times. Mean values with standard deviations were reported. Means were compared by one way ANOVA and Tukey post hoc was applied to check the significant differences among means. The computer software used in this study was SPSS (version10.1, 2000, SPSS Inc., Chicago, IL, USA).

### 3.8. Isolation and Sequencing of G. chorda DNA

DNA was extracted from *G. chorda* according to the CTAB DNA extraction method with some modifications [28]. First, the fresh sample was thoroughly washed with ultrapure water and freeze-dried. Then, approximately 10 mg dried sample was well ground into a powder and added to a 2.0 mL tube. The sample was suspended in 1 mL Tris-EDTA (TE) buffer, vortexed, and then placed on ice for 10 min. After that, the sample was centrifuged at 4 °C, 15,000× *g* for 1 min. The supernatant was collected in a new 2.0 mL tube containing 1% SDS and 0.1 mg/mL proteinase K and then incubated at 37 °C for 1 h. After the incubation, the sample containing 0.7 M NaCl and 1% CTAB was incubated at 65 °C for 10 min. Then the sample was added to an equal volume of phenol-chloroform-isoamyl alcohol, vortexed and centrifuged at 4 °C, 15,000× *g* for 5 min. Afterward, the supernatant (water-layer) was moved to a new 2.0 mL tube containing an equal volume of chloroform, vortexed and centrifuged at 4 °C, 15,000× *g* for 5 min. The supernatant (water-layer) was moved to a new 1.5 mL tube containing 0.6 volume of 2-propanol, mixed gently, centrifuged at 4 °C, 15,000× *g* for 5 min, and then discard the supernatant. The pellet was rinsed with 70% ethanol, centrifuged at 4 °C, 12,000× *g* for 5 min, then discard the supernatant and air-dried on a clean bench for 10 min. The DNA sample was dissolved in TE buffer containing 20 ng/mL RNase A in the final concentration approximately 100 ng/μL DNA concentration and incubate at 37 °C for 30 min. Finally, the concentration and purity of *G. chorda* DNA were analysed by NanoDrop 2000 Spectrometer (Thermo Fisher Scientific, Waltham, MA, USA).

The nucleotide sequences of the DNA were analysed by using a next generation sequencer, Ion PGM System (Thermo Fisher Scientific, Waltham, MA, USA). The data were assembled with CLC Genomics Workbench 9.5.4 (QIAGEN, Hilden, Germany). The gap in PE-γ gene was filled by Sanger sequencing. Nucleotide and deduced amino acid sequences of the PE subunits from *G. chorda* were aligned using EMBL-EBI Clustal Omega [29]. Molecular weight and isoelectric point of *G. chorda* phycobiliproteins were calculated from deduced amino acid sequences by using the compute pI/Mw tool, ProtParam (https://web.expasy.org/protparam/, accessed on 5 December 2022).

### 3.9. In Silico Analysis

#### 3.9.1. Building 3D Structure of *G. chorda* Phycobiliproteins and RuBisCo

The 3D structures of *G. chorda* phycobiliproteins and RuBisCo were predicted using SWISS-MODEL server (http://swissmodel.expasy.org/); the predicted structure was output by PyMOL version 2.5.2 (Schrödinger, LLC, NY, USA) [30], and the secondary structures were analysed by PDBsum webserver [31].

#### 3.9.2. Ligand Preparation

The peptides IDHY and LVVER were found potential inhibitors of ACE and selected for *molecular docking* analysis. Their 3D structure was built in Biovia Discovery Studio Client v19.1.0.18287 (Dassault Systèmes, San Diego, CA, USA), and energy was minimized. The peptides were then prepared for the flexible docking software using AutoDock tools version 1.5.7.

#### 3.9.3. Protein Preparation

The appropriate 3D structures of ACE (identity 1O86) in complex with Lisinopril were retrieved from Protein Data Bank (https://www.rcsb.org/), as well as the structure of ACE with Captopril (1UZF). All the ligands were removed using Biovia Discovery Studio v19.1 software and the protein was prepared for docking using AutoDock tools and Kollman charges were used.

#### 3.9.4. Grid Generation and Molecular Docking

The grid was centered on the crystallized ligand in the crystal structure of ACE using AGFR 1.0 tool, dedicated software for flexible docking preparation. The dimensions of the box were: 32 Å × 32 Å × 32 Å, center-x = 40.915 Å, center-y = 39.340 Å, center-z = 42.338 Å. Docking was performed at 2 × 10^6^ evaluations in 6 runs each using AutoDockFR software. Results were analysed using PyMOL version 2.5.2, Biovia Discovery Studio and LigPlot plus version 2.2.5 (Cambridgeshire, UK) for the 2D interaction diagram.

## 4. Conclusions

It has been demonstrated in this study that *G. chorda* WSP contained a high amount of PE, but also PC, APC and RuBisCo. Their hydrolysis by thermolysin produced bioactive peptides with DPP-IV inhibitory and DPPH scavenging activities, and particularly ACE inhibitory activity. Fractionation and identification of peptides in the thermolysin hydrolysate produced three peptides with high potential for ACE inhibition and among them two new peptides. Molecular docking analysis of the interaction between the new peptide IDHY and ACE revealed it was a promising inhibitor. In addition, the 3D structures of phycobiliproteins and RuBisCo were predicted, and the location of bioactive peptides with ACE inhibitory activity was mapped to the structure. The structural information surrounding bioactive peptides in the protein structure such as solvent accessible area, the type of secondary conformation and solubility was then determined, to understand the structure–function relationship of *G. chorda* WSP. From these results, it could be expected that *G. chorda* protein hydrolysates finds application as an ingredient in the prevention of hypertension, and bioactive peptides as a component in functional foods.

## Figures and Tables

**Figure 1 marinedrugs-21-00049-f001:**
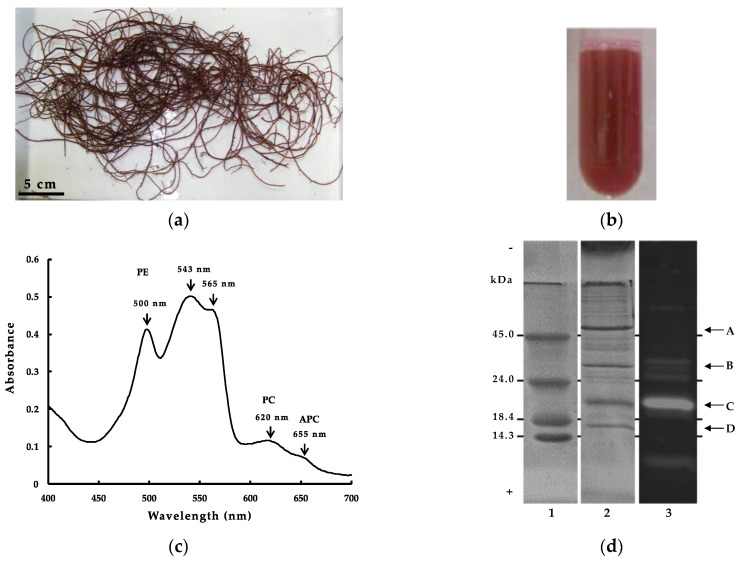
(**a**) Harvested *G. chorda* on the coast of Hakodate; (**b**) *G. chorda* WSP; (**c**) Visible ray absorption spectrum of *G. chorda* WSP; (**d**) SDS-PAGE electrophoresis of *G. chorda* WSP: (**1**) MW markers; (**2**) *G. chorda* WSP stained by Coomassie Brilliant Blue R-250; (**3**) *G. chorda* WSP fluorescent photography; (A) RuBisCo-L; (B) PE-γ; (C) PE-α, PE-β; (D) RuBisCo-S.

**Figure 2 marinedrugs-21-00049-f002:**
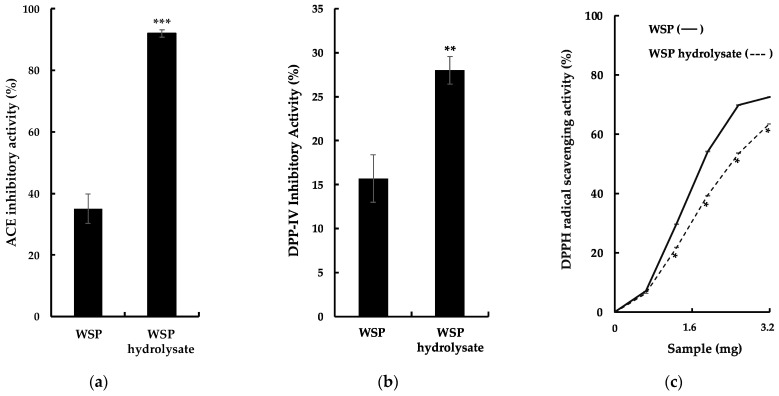
Bioactive properties of *G. chorda* WSP and its thermolysin hydrolysate: (**a**) ACE inhibitory activity; (**b**) DPP-IV inhibitory activity; (**c**) DPPH scavenging activity. *: *p* < 0.05; **: *p* < 0.01; ***: *p* < 0.001.

**Figure 3 marinedrugs-21-00049-f003:**
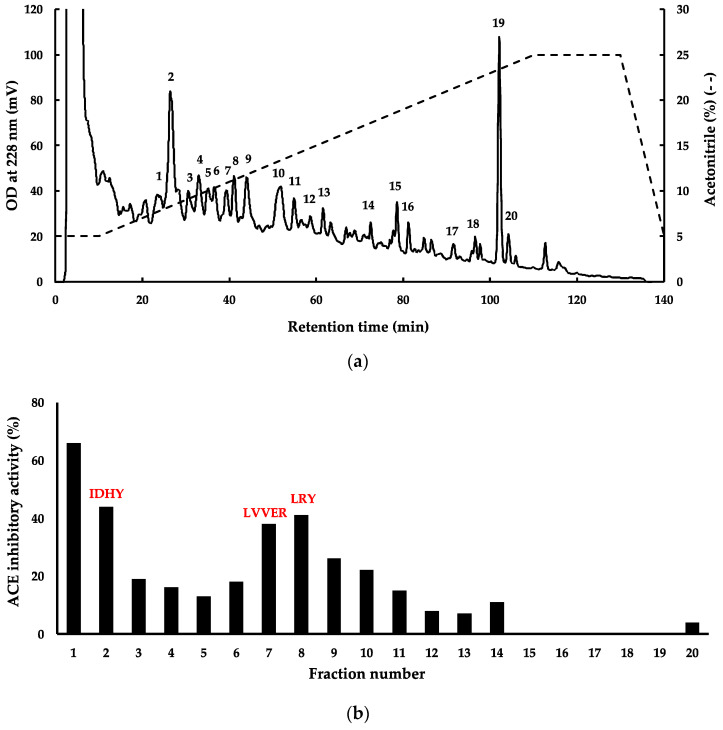
(**a**) Isolation of ACE inhibitory peptides from *G. chorda* WSP hydrolysate by RP-HPLC; (**b**) evaluation of ACE inhibitory activity of the pooled fractions 1–20.

**Figure 4 marinedrugs-21-00049-f004:**
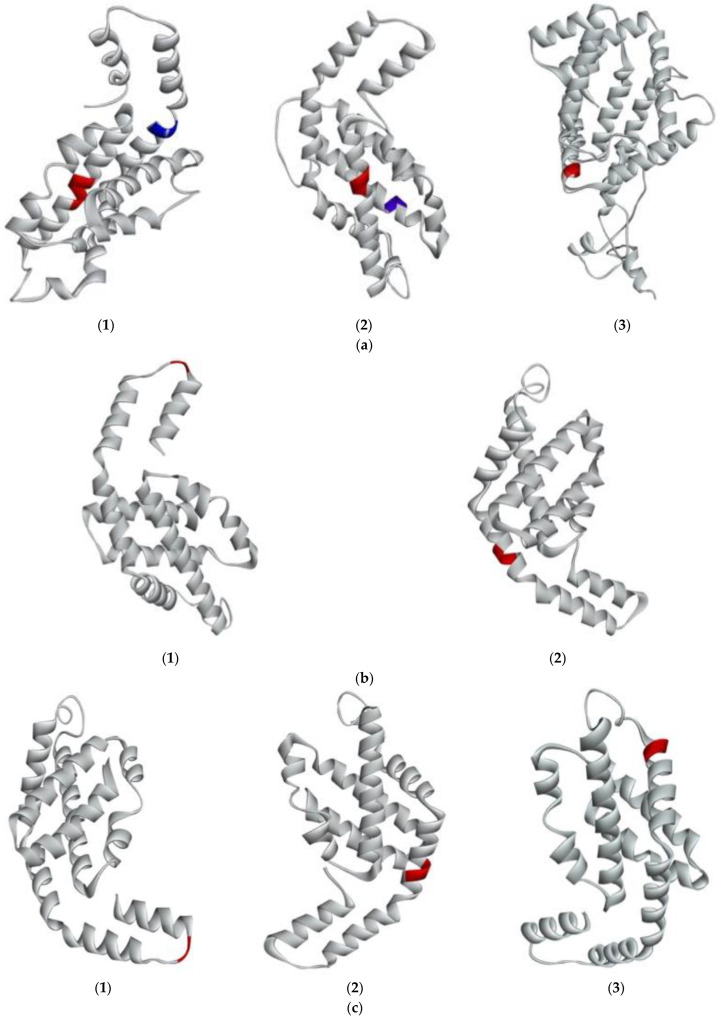
Predicted 3D structure of G. chorda phycobiliproteins and RuBisCo, and location of bioactive peptides. (**a**) (**1**)PE-α showing IDHY (red) and AR (blue); (**2**) PE-β showing LRH (red); (**3**) PE-γ showing VR (red); (**b**) (**1**) PC-α showing LEE (red); (**2**) PC-β showing AR (red); (**c**) (**1**) APC-α showing AR (red); (**2**) APC-β showing VR (red); (**3**) APC-**γ** showing FQ (red); (**d**) (**1**) RuBisCo-L showing LVVER (red); IW (blue); VK (green); (**2**) RuBisCo-S.

**Figure 5 marinedrugs-21-00049-f005:**
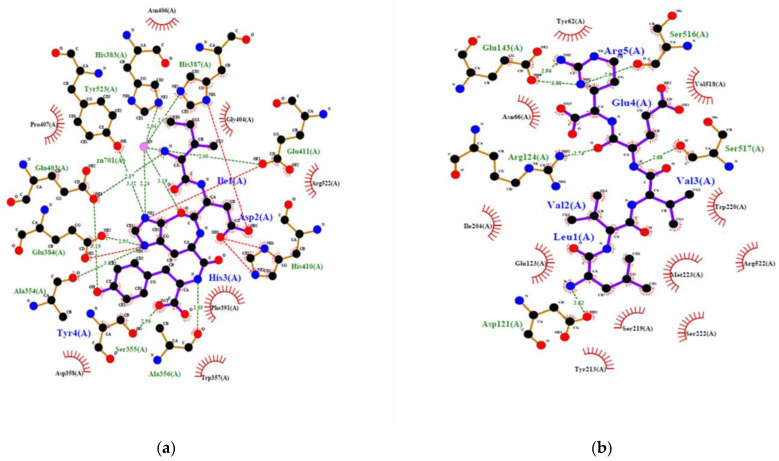
2D view of the interaction of human ACE with ACE inhibitory peptides: (**a**) IDHY; (**b**) LVVER; (**c**) Lisinopril; (**d**) Captopril.

**Table 1 marinedrugs-21-00049-t001:** Bioactive peptides found in *G. chorda* soluble proteins following hydrolysis by thermolysin.

							Properties			Secondary Structure ^a^
	Sequence	Occurrence	Position	IC_50_ (µM)	Proteins	[M+H]^+^	Solubility ^b^	SAS in nm^2 c^	Peptide Ranker Score ^d^	
Detected	IDHY	1	86–89	-	PE-α	547.25	Good	0.381	0.285	Helix
	LRY	1	90–92	5.1	PE-β	451.26	Good	0.486	0.512	Helix
	LVVER	1	159–163	-	RuBisCo-L	615.35	Good	0.316	0.044	Helix
Potential ^e^	AR	1/1/1	36–37, 56–57, 15–16	96	PE-α/ PC-β/ APC-α	245.14	Good	0.125 0.700 0.865	0.394	Helix Helix -
	VR	1/1	164–165, 38–39	52.8	PE-γ/ APC-β	273.17	Good	0.579 1.091	0.115	Helix Helix
	LEE	1	115–117	100	PC-α	389.17	Good	0.809	0.0350	Helix
	FQ	1	59–60	51.23	APC-γ	293.13	Poor	0.843	0.916	Helix
	IW	1	285–286	4.7	RuBisCo-L	317.16	Poor	0.453	0.944	Helix
	VK	1	134–135	13.0	RuBisCo-L	245.16	Good	0.662	0.033	Beta strand

^a^ PDBSUM was used to generate the secondary structure of peptides in the protein. ^b^ Solubility was predicted by INNOVAGEN peptide calculator (https://pepcalc.com/peptide-solubility-calculator.php, accessed on 10 November 2022). ^c^ Solvent accessible surface (SAS) for each residue was calculated by the SASA script in Gromacs. ^d^ Predicted score by PeptideRanker (http://distilldeep.ucd.ie/PeptideRanker/, accessed on 14 November 2022).^e^ Potential bioactive peptides with IC_50_ < 100 µM were considered. -: Non defined.

**Table 2 marinedrugs-21-00049-t002:** Secondary structure in G. chorda WSP ^a^.

	PE			PC		APC			RuBisCo	
	α	β	γ	α	β	α	β	γ	L	S
Secondary structure descriptors ^b^										
helices	10 (74.4%) ^c^	10 (74.9%)	14 (62.1%)	9 (76.6%)	10 (71.2%)	9 (78.1%)	7 (78.3%)	9 (77.5%)	25 (42.2%)	3 (20.3%)
beta hairpins	-	-	-	-	-	-	-	-	2	2
beta sheets	-	-	-	-	-	-	-	-	3	2
beta strands	-	-	-	-	-	-	-	-	15 (14.8%)	4 (40.5%)
beta bulges	-	-	-	-	-	-	-	-	3	2
Helix–helix interactions	18	17	14	17	18	19	13	20	22	-
beta turns	5	4	13	2	4	6	7	6	27	5
gamma turns	1	-	1	-	-	-	-	-	3	-

^a^ Secondary structure was generated by PDBsum (https://www.ebi.ac.uk/thornton-srv/databases/pdbsum/Generate.html, accessed on 5 December 2022). ^b^ Number of secondary structures. ^c^ Rate of amino acid residues involved in secondary structure formation among total amino acid residues. -: Non defined.

**Table 3 marinedrugs-21-00049-t003:** Molecular docking results of the interaction of IDHY and LVVER from G. chorda against crystal structure of ACE.

Ligand	Number Clusters	Affinity (kcal/mol) ^a^	RMSDi in A	Number of Bonds with Zn^2+^	Hydrogen Bonds ^b^			Number of Non-Bonded Contacts
					Number	Residues	Length (Å)	
IDHY	10	−9.50	0.0	2	7	TYR523 GLU403 GLU403 GLU384 ALA356 SER355 ALA354	3.323 3.189 2.568 2.926 2.847 2.958 3.067	102
LVVER	10	−6.9	0.0	0	6	SER517 SER516 GLU143 GLU143 ARG124 ASP121	2.681 2.994 2.998 2.941 2.741 2.619	62
Lisinopril	-	-	-	2	8	TYR523 TYR520 HIS513 LYS511 HIS387 ALA354 HIS353 HIS353	2.775 2.556 3.109 2.934 3.327 2.917 2.760 3.241	48
Captopril	-	-	-	1	5	TYR520 HIS513 LYS511 HIS353 GLN281	2.657 2.694 2.730 2.542 3.074	36

^a^ Affinity energy was calculated by AutoDockFR suite suite version 1.0 (The Scripps Research Institute, Centre For Computational Structural Biology, California, USA). The hit with the lower energy was considered. ^b^ Number of hydrogen bonds and non-bonded contact was calculated by LigPlotPlus software and Discovery Studio using default parameters. -: Non defined.

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
