# Peer review of "Characterisation of Bioactive Peptides from Red Alga Gracilariopsis chorda"

_marinedrugs, 2023, doi:10.3390/md21010049_

Round 1

Reviewer 1 Report

This is an interesting study of ACE inhinitory activity in the thermolysin hydrolysate of red algae WSP which has been furtehr investigated to determine molecular emchanisms of action of 2 isolated peptides. The comments on this manuscript are stated below:

1. Please describe what is 'dulse' (i.e., a tpe of red algae) and its scientific name the very first time it is mentioned in the manuscript.

2. Please elucidate upon what is a 'plastid' and why it is significant

3. The terms "peptides" and 'hydroltsate' have been used interchangeably which causes confusion to the reader. Please use 'petides' only when a specific mix of two or more identified peptides is being mentioned. Inb all other cases, the term 'hydrolysate' is more appropriate. For example, in figure 2, the columns should be labelled as "WSP" and "WSP hydrolysate" instead of 'proteins' and 'peptides'. This is critically important to avoid any misundertsandings.

4. The authors have focused on detemrining the molecular emchanisms of 2 identified peptides, namley IDHY and LVVER. However, no data is presented to show that these 2 peptides actually had any signifciant ACE inhibitor properties. Please conduct these assays using synthetic peptides (LVVER and IDHY) to demonstrate their ability to inhibit ACE in the 'real world'. Without this critical piece of evidence, the molecular docking and otehr in silico studies cannot be justified.

5. It is starnge that the authors needed to investigate the genetic sequence instead of simply studying the protein sequence and seconary/tertiary structures of PE-gamma. A stronger rationale for these studies shoudl be provided.

6. Finally, many studies have been published using enzymatic hydrolysis of food derived proteins to genrate ACE inhibitor peptides, and further studies have used synthetic analogues of such peptides to examine the molecular mechanisms of such. The authors are strongly advised to mention and cite such studies of food derived peptides with ACE inhibitor actions in the Introduction section.

Author Response

Reviewer 1

Comments and Suggestions for Authors

This is an interesting study of ACE inhibitory activity in the thermolysin hydrolysate of red algae WSP which has been further investigated to determine molecular mechanisms of action of 2 isolated peptides. The comments on this manuscript are stated below:

âž” We appreciate helpful comments and suggestions from Reviewer 1. We accepted these comments and revised the manuscript. Following are our replies, and we hope that Reviewer 1 kindly evaluates our responses. The parts, which were added or modified in the manuscript, are red letters in the manuscript.

  1. Please describe what is 'dulse' (i.e., a type of red algae) and its scientific name the very first time it is mentioned in the manuscript.

âž” Thank you very much for your comment and suggestion.

We added a description of dulse and its scientific name and changed the sentences as follows: “In a previous study, we analyzed the steric structure of PE from Japanese dulse (Deverolaea inkyureii), which is a type of red algae distributed in frigid waters. We found that its structure is an α-helix-rich (αβ)6 hexamer complex with a toroidal shape [9].” (Lines 70-73)

Accompanied with this correction, “dulse and Pyropia pseudolinearis” was revised to “red algae”. (Line 56)

  1. Please elucidate upon what is a 'plastid' and why it is significant

âž” Thank you very much for your comment.

We changed “plastid” to “chloroplast” to unify terminology in the manuscript. The major WSPs in red algae are phycobiliproteins and RuBisCo. The genes for those proteins are encoded in chloroplast DNA, so it is very important to focus on chloroplasts.

Therefore, we revised the sentences to “To optimize the application of G. chorda WSP in the preparation of hydrolysate with high ACE inhibitory activity by in silico analysis, it is important to elucidate the structure of WSP. However, G. chorda contained relatively amounts of polysaccharides. In addition, the phycobiliproteins are very hydrophobic and adsorb to the column resin, resulting in very low yields. For these reasons, it was very difficult to isolate and purify the major proteins of WSP. Therefore, we needed to approach the elucidation of their primary structure by genetic analysis. Since the genes of phycobiliproteins and RuBisCo, major proteins of WSP, are encoded in chloroplast DNA, as in previous studies [4,20], we determined the complete chloroplast genome and manually annotated it.” (Lines 200-208)

  1. The terms "peptides" and 'hydrolysate' have been used interchangeably which causes confusion to the reader. Please use 'peptides' only when a specific mix of two or more identified peptides is being mentioned. In all other cases, the term 'hydrolysate' is more appropriate. For example, in figure 2, the columns should be labeled as "WSP" and "WSP hydrolysate" instead of 'proteins' and 'peptides'. This is critically important to avoid any misunderstandings.

➔Thank you very much for your comment and suggestion.

We revised the labels in Figure 2.

  1. The authors have focused on determining the molecular mechanisms of 2 identified peptides, namely IDHY and LVVER. However, no data is presented to show that these 2 peptides actually had any significant ACE inhibitor properties. Please conduct these assays using synthetic peptides (LVVER and IDHY) to demonstrate their ability to inhibit ACE in the 'real world'. Without this critical piece of evidence, molecular docking and other in silico studies cannot be justified.

âž” Thank you very much for your comment.

We revised the sentences to “In the previous study, we identified a novel ACE inhibitory peptide LRY from dulse [2]. The IC50 value of synthesized LRY against the ACE from rabbit lung was 0.044 μmol, and its effect was the same level as that of the sesame peptide LVY (IC50 value: 0.045 μmol) used as a Food for Specified Health Uses (FOSHU) in Japan. Then, we performed a docking simulation of LRY with human ACE and found that - 9.50 kcal/mol. Since the affinity of IDHY for human ACE in this study was - 9.50 kcal/mol, it was presumed that IDHY has a hypotensive effect on humans as well as LVY. We are currently planning a clinical trial on the blood pressure-lowering effects of LRY and IDHY from red algae in collaboration with a company.” (Lines 177-185)

Accompanied with this correction, “[13]” was revised to “[3,20]”. (Line 177)

  1. It is strange that the authors needed to investigate the genetic sequence instead of simply studying the protein sequence and secondary/tertiary structures of PE-gamma. A stronger rationale for these studies should be provided.

âž” Thank you very much for your comment and suggestion.

For in silico analysis, it was necessary to know the primary structure of phycobiliproteins and RuBisCo, the main components of WSP. However, G. chorda contains relatively amounts of polysaccharides, making it very difficult to isolate and purify the proteins.

Therefore, we revised the sentences to  “To optimize the application of G. chorda WSP in the preparation of hydrolysate with high ACE inhibitory activity by in silico analysis, it is important to elucidate the structure of WSP. However, G. chorda contained relatively amounts of polysaccharides. In addition, the phycobiliproteins are very hydrophobic and adsorb to the column resin, resulting in very low yields. For these reasons, it was very difficult to isolate and purify the major proteins of WSP. Therefore, we needed to approach the elucidation of their primary structure by genetic analysis. Since the genes of phycobiliproteins and RuBisCo, major proteins of WSP, are encoded in chloroplast DNA, as in previous studies [4,20], we determined the complete chloroplast genome and manually annotated it.” (Lines 200-208)

In addition, it is noteworthy that we found the contig containing PE-γ gene encoded in nuclear DNA by next generation sequencing.

Therefore, we revised the sentences to “The α and β subunits of PE construct (αβ)6 as described above. On the other hand, PE-γ is placed in the central space of the (αβ)6 hexamer complex as a linker protein to form part of the rod of the phycobilisome. Compared to the α and β subunits, there have been fewer structural and functional studies of the γ subunits, because the gene is encoded in the nucleus [21].” (Lines 213-217)

  1. Finally, many studies have been published using enzymatic hydrolysis of food derived proteins to generate ACE inhibitor peptides, and further studies have used synthetic analogs of such peptides to examine the molecular mechanisms of such. The authors are strongly advised to mention and cite such studies of food derived peptides with ACE inhibitor actions in the Introduction section.

âž” Thank you very much for your comment and suggestion.

We added the sentences “Particularly, new ACE inhibitory peptides were purified from algae WSP hydrolyzed by thermolysin [10], and inhibition mechanism was found competitive and non-competitive [10,11]. (Lines 76-78)

Reviewer 2 Report

The authors reported about bioactive peptides from one red algae. The subject is new and of great interest in the scientific community. I suggest some modifications to improve the MS:

Title: to remove the word “prepared”

To substitute the word “Rubulose” by “ribulose”

 I suggest the quantification of phycobiliproteins and RuBisCo, to prove that they are in higher quantity in wsp

Page 2, line 53. To put the scientific name of “dulse”

Topic 3.2. What is the purpose of DNA isolation? Confirm the species of red algae?

It is not clear which DNA sequence was analyzed. The sequence of PE, PC, APC and Rubisco? What primes are used? What gene was sequenced?

Page 13. Line 312. the sentence “ The visible ray absorption spectrum of dulse protein extracts analyzed by a spectrophotometer” is confusing. What was analyzed? Protein quantification?

It needs to explain how the proteins were extracted.

Why was it not quantifying PE, PC and APC?

To show antioxidant activity, it is ideal to use one more methodology. The use of only one antioxidant methodology does not mean that it has no activity.

Page 2, line 90. The authors should follow the same order of material and methods.

Chromophore absorbances should be in materials and methods.

Based on what the authors state “high content of PE in the mixture”. What is the standard to be considered high?

The sentence “Similar results were obtained for other red algae such as dulse” is very vague.

Page 3. Line 125. To add the value of DPPH antioxidant activity.

The authors infer that the peptides are present in phycobilins and rubisco. However, hydrolysis was not performed on these purified proteins. In the protein extract there are several proteins that after hydrolysis can also produce the peptides found.

To format all tables

Table 2 should be as supplementary material.

Author Response

Reviewer 2

Comments and Suggestions for Authors

The authors reported about bioactive peptides from one red algae. The subject is new and of great interest in the scientific community. I suggest some modifications to improve the MS: 

âž” We appreciate helpful comments and suggestions from Reviewer 2. We accepted these comments and revised the manuscript. The followings are our replies, and we hope that Reviewer 2 kindly evaluates our responses. The parts, which were added or modified in the manuscript, are red letters in the manuscript.

Title: to remove the word “prepared”

âž” Thank you very much for your suggestion.

We revised the title to “Characterisation of bioactive peptides from red alga Gracilariopsis chorda“ (Line 2)

To substitute the word “Rubulose” by “ribulose”

âž” Thank you for your comment.

We revised “Rubulose” to “Ribulose” (Line 21)

 I suggest the quantification of phycobiliproteins and RuBisCo, to prove that they are in higher quantity in wsp.

âž” Thank you for your suggestion.

  1. chorda contained relatively amounts of polysaccharides. In addition, the phycobiliproteins are very hydrophobic and adsorb to the column resin, resulting in very low yields. For these reasons, it was very difficult to isolate and purify the major proteins of WSP. Hence, we could not quantify them.

Therefore, we calculated the ratio of PE, PC, and APC present based on the absorbance of the visible light absorption spectrum: i.e., the concentration of each phycobiliprotein was calculated from the absorbance at 562 nm, 652 nm, and 615 nm using the following formula given by Bennett and Bogorad [9] (PE = 5.0 mg/mL, PC = 1.8 mg/mL, APC = 0.01 mg/mL). (Lines 114-117)

Page 2, line 53. To put the scientific name of “dulse”

âž” Thank you very much for your suggestion.

We added a description of dulse and its scientific name and changed the sentences as follows: “In a previous study, we analyzed the steric structure of PE from dulse (Deverolaea inkyureii), which is a type of red algae harvested in cold waters of Japan. We found that its structure is an α-helix-rich (αβ)6 hexamer complex with a toroidal shape [9].” (Lines 70-73)

Accompanied with this correction, “dulse and Pyropia pseudolinearis” was revised to “red algae”. (Line 56)

Topic 3.2. What is the purpose of DNA isolation? Confirm the species of red algae?

âž” Thank you very much for your comments.

The major WSP in red algae are phycobiliproteins and RuBisCo. The genes for those proteins are encoded in chloroplast DNA, so it is very important to focus on chloroplasts.

Therefore, we revised the sentences to “To optimize the application of G. chorda WSP in the preparation of hydrolysate with high ACE inhibitory activity by in silico analysis, it is important to elucidate the structure of WSP. However, G. chordacontained relatively amounts of polysaccharides. In addition, the phycobiliproteins are very hydrophobic and adsorb to the column resin, resulting in very low yields. For these reasons, it was very difficult to isolate and purify the major proteins of WSP. Therefore, we needed to approach the elucidation of their primary structure by genetic analysis. Since the genes of phycobiliproteins and RuBisCo, major proteins of WSP, are encoded in chloroplast DNA, as in previous studies [4,20], we determined the complete chloroplast genome and manually annotated it.” (Lines 200-208)

It is not clear which DNA sequence was analyzed. The sequence of PE, PC, APC and Rubisco? What primes are used? What gene was sequenced?

âž” Thank you for your comments.

As described in 3.8. Isolation and sequencing of G. chorda DNA, we analyzed DNA from G. chorda by next-generation sequencing and Sanger sequencing and determined the nucleotide sequences of PE-α, PE-β, PE-γ, PC-α, PC-β, APC-α, APC-β, Rubisco-L and Rubisco-S genes. (Lines 403-431)

Page 13. Line 312. the sentence “ The visible ray absorption spectrum of dulse protein extracts analyzed by a spectrophotometer” is confusing. What was analyzed? Protein quantification?

âž” Thank you very much for your comment.

              As mentioned above, G. chorda contained relatively amounts of polysaccharides. In addition, the phycobiliproteins are very hydrophobic and adsorb to the column resin, resulting in very low yields. For these reasons, it was very difficult to isolate and purify the major proteins of WSP. Hence, we could not quantify them.

Therefore, we calculated the ratio of PE, PC, and APC present based on the absorbance of the visible light absorption spectrum: i.e., the concentration of each phycobiliprotein was calculated from the absorbance at 562 nm, 652 nm, and 615 nm using the following formula given by Bennett and Bogorad [9] (PE = 5.0 mg/mL, PC = 1.8 mg/mL, APC = 0.01 mg/mL). (Lines 114-117)

It needs to explain how the proteins were extracted.

âž” Thank you for your suggestion.

We added the sentences “The WSP from G. chorda were prepared according to the same method as in our previous paper [3]: i.e., frozen G. chorda was lyophilized and ground into a fine powder. To this powder, 20 v/w distilled water was added and proteins were extracted at 4°C for 7 hours. The extract was centrifuged at 4oC, 15,000 g for 10 min, and then the supernatant was used as G. chorda WSP. The visible ray absorption spectrum of G. chorda WSP was analyzed by a spectrophotometer (UV-1800, Shimadzu, Kyoto, Japan).” (Lines 331-336)

Why was it not quantifying PE, PC and APC?

âž” Thank you for your suggestion.

              As mentioned above, G. chorda contained relatively amounts of polysaccharides. In addition, the phycobiliproteins are very hydrophobic and adsorb to the column resin, resulting in very low yields. For these reasons, it was very difficult to isolate and purify the major proteins of WSP. Hence, we could not quantify them.

Therefore, we calculated the ratio of PE, PC, and APC present based on the absorbance of the visible light absorption spectrum: i.e., the concentration of each phycobiliprotein was calculated from the absorbance at 562 nm, 652 nm, and 615 nm using the following formula given by Bennett and Bogorad [9] (PE = 5.0 mg/mL, PC = 1.8 mg/mL, APC = 0.01 mg/mL). (Lines 114-117)

To show antioxidant activity, it is ideal to use one more methodology. The use of only one antioxidant methodology does not mean that it has no activity.

âž” Thank you for your comment.

              In a study of Japanese dulse, it was reported that the antioxidant activity of WSP (DPPH and ABTS radical scavenging activity) is relatively low and that the main antioxidant activity of WSP is derived from the chromophore of phycobiliproteins [6]. Therefore, we believe that the antioxidant activity of WSP and its hydrolysates is low, even when antioxidant activity is measured by other methods. (Lines 147-151)

Page 2, line 90. The authors should follow the same order of material and methods.

âž” Thank you for your suggestion.

              The order of Materials and Methods and the order of Results and Discussion have been unified. (Lines 330, 347, 362, 377, 388, 398, 403, 433, 438, 443, 449)

Chromophore absorbances should be in materials and methods.

âž” Thank you for your suggestion.

We added the sentence “The visible ray absorption spectrum of G. chorda WSP was analyzed by a spectrophotometer (UV-1800, Shimadzu, Kyoto, Japan).” (Lines 335-336)

Based on what the authors state “high content of PE in the mixture”. What is the standard to be considered high?

âž” Thank you for your comment.

              As mentioned above, G. chorda contained relatively amounts of polysaccharides. In addition, the phycobiliproteins are very hydrophobic and adsorb to the column resin, resulting in very low yields. For these reasons, it was very difficult to isolate and purify the major proteins of WSP. Hence, we could not quantify them.

Therefore, we calculated the ratio of PE, PC, and APC present based on the absorbance of the visible light absorption spectrum: i.e., the concentration of each phycobiliprotein was calculated from the absorbance at 562 nm, 652 nm, and 615 nm using the following formula given by Bennett and Bogorad [9] (PE = 5.0 mg/mL, PC = 1.8 mg/mL, APC = 0.01 mg/mL). (Lines 114-117)

The sentence “Similar results were obtained for other red algae such as dulse” is very vague.

âž” Thank you for your comment.

We added the sentence “Similar results were obtained for other red algae such as dulse [3]: i.e., we calculated the ratio of PE, PC, and APC present based on the absorbance of the visible light absorption spectrum, and the main component of the phycobiliprotein was PE (PE = 5.0 mg/mL, PC = 1.8 mg/mL, APC = 0.01 mg/mL) [14].” (Lines 113-117)

Page 3. Line 125. To add the value of DPPH antioxidant activity.

âž” Thank you for your comment.

We added the sentences “(63% DPPH scavenging activity at 3.2mg).” (Line 142)

The authors infer that the peptides are present in phycobilins and rubisco. However, hydrolysis was not performed on these purified proteins. In the protein extract there are several proteins that after hydrolysis can also produce the peptides found.

âž” Thank you for your comment.

              As mentioned above, G. chorda contained relatively amounts of polysaccharides. In addition, the phycobiliproteins are very hydrophobic and adsorb to the column resin, resulting in very low yields. For these reasons, it was very difficult to isolate and purify the major proteins of WSP. Hence, we could not quantify them. Therefore, we carried out to approach the elucidation of their primary structure by genetic analysis.

To format all tables

âž” Thank you for your comment.

The format of the manuscript, including Table 2, has been revised by the Editorial Office.

Table 2 should be as supplementary material.

âž” Thank you for your suggestion.

              We moved Table 2 to the supplemental material as Table S1. (Lines 472-473)

Accompanied with this correction, “Table 3” and Table 4” were revised to “Table 2” and Table 3”. (Lines 241, 244, 265, 270, 283)

Reviewer 3 Report

The present study “Characterisation of bioactive peptides prepared from red alga Gracilariopsis chorda” is valuable for the reader and has practical significance. However, there are some other points that should be considered before it publications.

·         Lines 40-42: please add reference for this statement.

·         I would suggest to write some medicinal use of red alga with reference.

·         Objectives should be written more in details.

·         The study lacks in-depth literature/discussion of previous studies. Authors are strongly advised to add more previous studies and compared with previous studies.

·         Authors should write some practical applicability of this study in conclusion section.

Author Response

Reviewer 3

The present study “Characterisation of bioactive peptides prepared from red alga Gracilariopsis chorda” is valuable for the reader and has practical significance. However, there are some other points that should be considered before it publications.

âž” We appreciate helpful comments and suggestions from Reviewer 3. We accepted these comments and revised the manuscript. The followings are our replies, and we hope that Reviewer 3 kindly evaluates our responses. The parts, which were added or modified in the manuscript, are red letters in the manuscript.

  • Lines 40-42: please add reference for this statement. 

➔Thank you very much for your suggestion.

We added reference [1]. (Line 44)

  • I would suggest to write some medicinal use of red alga with reference.

➔Thank you for your suggestion.

We added sentence “In this regard, bioactive metabolites found in algae are used to treat AIDS (acquired immune-deficiency syndrome), inflammation, arthritis, and microbial infections [2]”. (Lines 45-47)

  • Objectives should be written more in details.

➔Thank you for your suggestion.

We added sentence “In this regard, combining information from the chloroplast protein structure and bioactive properties of protein hydrolysates, could be important to understand structural factors that influence the release of bioactive peptides from red algae chloroplast proteins.” (Lines 91-94)

  • The study lacks in-depth literature/discussion of previous studies. Authors are strongly advised to add more previous studies and compared with previous studies. 

➔Thank you for your suggestion.

We added sentences “Proteins from the red alga dulse previously showed low radical scavenging activity, probably due to low content of amino acids capable of scavenging free radicals [6]. In addition, high ACE inhibitory activity was also observed in red alga Porphyra dioica proteins hydrolyzed by Alcalase and Flavourzyme [15], and DPP-IV inhibitory effect was noticed in Irish dulse proteins digested by Corolase PP [16]. In a study of Japanese dulse, it was reported that the antioxidant activity of WSP (DPPH and ABTS radical scavenging activity) is relatively low and that the main antioxidant activity of WSP is derived from the chromophore of phycobiliproteins [6].” (Lines 91-94)

  • Authors should write some practical applicability of this study in conclusion section.

➔Thank you for your suggestion.

We added sentence “From these results, it could be expected that G. chorda protein hydrolysates find application as an ingredient in the prevention of hypertension, and bioactive peptides as a component in functional foods.” (Lines 468-470)

Round 2

Reviewer 1 Report

The authors have successfully addressed the concerns raised.

Reviewer 2 Report

The authors improved the MS.